# The Use of Different Anthropometric Indices to Assess the Body Composition of Young Women in Relation to the Incidence of Obesity, Sarcopenia and the Premature Mortality Risk

**DOI:** 10.3390/ijerph191912449

**Published:** 2022-09-29

**Authors:** Martina Gažarová, Maroš Bihari, Marta Lorková, Petra Lenártová, Marta Habánová

**Affiliations:** 1Institute of Nutrition and Genomics, Faculty of Agrobiology and Food Resources, Slovak University of Agriculture in Nitra, Trieda Andreja Hlinku 2, 949 76 Nitra, Slovak Republic; 2AgroBioTech Research Centre, Slovak University of Agriculture in Nitra, Trieda Andreja Hlinku 2, 949 76 Nitra, Slovak Republic

**Keywords:** a body shape index, anthropometric indices, fat mass, muscle mass, obesity, female

## Abstract

The objective of the study was to evaluate the stratification of young women based on the assessment of body composition according to several currently recommended anthropometric indices and parameters, as well as the presence of obesity, sarcopenic obesity and the risk of premature death. Three hundred and three young Caucasian women aged 18–25 years were included in the cross-sectional observational study. For the purposes of the study, we used the bioelectrical impedance method and applied the obtained data to calculate indices defining obesity, sarcopenic obesity and premature mortality risk (ABSI z-score). We found significant differences between indicators of total and abdominal obesity when determining the rate of risk of premature death and diagnosis of obesity. Our results also suggest that FMI and FM/FFM indices correlate excellently with fat mass and visceral adipose tissue, better than BMI. Even in the case of abdominal obesity, FMI appears to correlate relatively strongly, more so than BMI. The results of the study support the opinion that in the assessment of body composition and health status, the presence of obesity (sarcopenic obesity) and the risk of premature death, anthropometric parameters and indices focusing not only on body weight (BMI, ABSI), but also on the proportionality and distribution of fat (WC, WHR, WHtR, VFA) and muscle tissue (FFMI, SMMI, FM/FFM ratio) should be used.

## 1. Introduction

Obesity is defined as excessive deposition of body fat or an excess of total body fat expressed through a body mass index of ≥30 kg/m^2^ [1]. According to the World Health Organization [2], overweight and obesity increase the risk of developing diabetes mellitus type 2, hypertension, ischemic heart disease or stroke, impaired glucose tolerance, degenerative diseases of the musculoskeletal system and many other health complications. Central obesity is an excess accumulation of fat in the abdominal area that wraps around abdominal organs and poses metabolic risks. To detect central obesity, measuring waist circumference is the easiest way and one of the most accurate anthropometric indicators of abdominal fat. Visceral adiposity is associated with higher metabolic morbidity and mortality due to an associated higher level of inflammatory processes compared to adiposity occurring elsewhere in the body [3]. Fat located in the abdominal area is associated with greater health risks than fat in peripheral areas, e.g., in the gluteal-femoral region [4,5,6].

Obesity can be assessed by direct measurement of body fat using computed tomography (CT), magnetic resonance imaging, DEXA, etc. [7]. However, these methods are expensive and have limitations that make their use in real clinical conditions difficult. Therefore, indirect indicators of obesity are used instead of them. Anthropometric measurements are simple, cheap and non-invasive [8]. The main components of anthropometric measurements are body weight and body height, waist circumference and hip circumference, limb circumference and skinfold thickness [9]. However, classical anthropometric methods have many limitations, and more accurate tools for estimating the content of adipose tissue are still being sought. The definition, diagnosis and classification of obesity based on BMI are increasingly questioned [10]. BMI has limited accuracy for predicting the amount and distribution of body fat and is unable to identify sarcopenic obesity [11]. Among other things, it does not sufficiently take into account gender, age and race [12]. Therefore, for the diagnosis of obesity and its risks, as well as the risk of metabolic disorders related to obesity, fat distribution indices are proposed, including the waist-to-height ratio (WHtR) and the ratio of waist circumference to hip circumference (WHR). In addition, there are new indices that are based on already existing indices, such as body shape index (ABSI) [13], body adiposity index (BAI) [14], body roundness index (BRI) [15], abdominal volume index (AVI) [14] and relative fat mass (RFM) [16]. Waist circumference and the waist-to-hip ratio, the so-called WHR index, specifically focus on the abdominal model of obesity. Furthermore, these parameters are limited by body constitution, gender, human race, variability in the denominator and others [17]. Waist-to-height ratio or WHtR showed a constant denominator when it comes to fixed height in adults and in studies achieved better results than BMI or WHR [18]. Krakauer and Krakauer [13] developed a body shape index (ABSI), which is based on waist circumference, body height and BMI. As the main factor in the calculation, it takes into account the waist circumference, which determines the roundness of the body. This index is supposed to limit the shortcomings of BMI, which does not differentiate the distribution of body fat and where the weight is concentrated on the body. ABSI is a more advanced index, as it distinguishes the distribution of body fat, whether it is concentrated around the waist, which is abdominal fat and at the same time riskier, or concentrated on the hips and buttocks, which is less risky from a health point of view. The ABSI was developed as a possible alternative to BMI and waist circumference [19]. As a predictor of mortality risk, ABSI outperformed the prediction of other indices and indicators, whether it is BMI, waist circumference, waist-to-height ratio, and waist-to-hip ratio. High ABSI values were shown to be associated with higher mortality rates much more than excess body mass index or waist circumference [20]. However, ABSI has limited clinical usefulness as it does not have cut-off points to identify individuals at high risk of obesity-related diseases [21,22]. Therefore, the ABSI z-score (log-transformation) has been designed that overcomes the limitations of ABSI. A meta-analysis by the authors Tian and Xu [23] showed that the ABSI z-score is the only measure of obesity related to sarcopenic obesity among all the obesity parameters that were assessed. A study by Chung et al. [22] examined the association of ABSI z-scores with both sarcopenic obesity and CVD and found that ABSI z-scores showed a better association with them compared to other obesity variables related to weight and WC. As mentioned above, BMI is not able to identify sarcopenic obesity [11]. Sarcopenic obesity is a syndrome characterized by a progressive and generalized loss of mass and skeletal muscle function, leading to adverse outcomes such as physical disability, poor quality of life and higher morbidity and mortality [24]. In the clinical setting, sarcopenic obesity is defined by higher fat mass (FM) compared to fat-free mass (FFM). Visceral adiposity may play a key role in the development of sarcopenic obesity [25].

Body composition can be affected by body water, fat or fat-free mass. In obese people, the proportion of water decreases and the proportion of fat in the body increases. Weight also changes with age, in children during growth and in adults as a result of fat accumulation. However, body weight taken without other measurements of body size is misleading as a person’s weight is highly dependent on height (i.e., tall people are generally heavier than short people). When monitoring changes in body weight, it is therefore necessary to monitor changes in the amount of muscle mass, water, body fat, basal metabolism, etc. [26,27].

The emergence and development of obesity is conditioned by various factors, from genetics to eating habits and an inactive lifestyle. However, correct diagnosis of obesity and interpretation of its nature and risk in relation to morbidity is very important and fundamental. In our study, we have evaluated various indirect anthropometric variables to confirm the usefulness and usability of other anthropometric markers of adiposity, similar to or even better than BMI in a group of young women. This study was also aimed to determine which anthropometric indices can be best associated with the premature mortality risk using ABSI z-score according to evaluation of body composition and to determine the presence of obesity, possibly sarcopenic obesity and excess fat.

## 2. Materials and Methods

### 2.1. Participants and Study Design

In the period between 2019 and 2021, we recruited young women from the university community for the purposes of the study. A group of 431 young adult female volunteers participated in a cross-sectional observational study. A written informed consent was obtained from all the participants prior to their involvement in the study. We used the following exclusion criteria for definitive participation in the study: age < 18 or >25 years; BMI > 50 kg/m^2^; pregnancy or presumption of pregnancy; professional athlete; presence of serious illnesses of a physical or psychological nature; contraindications for the bioimpedance measurement; increased physical activity just before the measurement; coffee; alcohol and fat intake ≤8 h prior to testing; and diuretic medications seven days prior to testing. One hundred and twenty-eight participants did not satisfy the criteria. Finally, 303 Caucasians were enrolled. The study was conducted according to the guidelines of the Declaration of Helsinki and approved by the Slovak University of Agriculture (SUA) in Nitra, Institute of Nutrition and Genomics, Slovakia; and by the Ethical Committee of the Specialized Hospital of St. Svorad Zobor in Nitra, Slovakia (Study No. 4/071220/2020).

### 2.2. Anthropometric Measurements

Body composition was analyzed by multi-frequency bioelectrical impedance analysis (MFBIA) using the device InBody 720 (Biospace Co. Ltd., Seoul, Korea). Participants were given information about the procedure and risks of BIA measurement in the case of an electrical device implanted in the body on the heart or in the case of pregnancy. Immediately before the body composition assessment, the participants were not allowed to drink a large volume of water or other fluids, consume alcohol for 24 h before testing, intake food high in sugar, salt or fat for 12 h prior to testing, and had to refrain from intense or strenuous physical activity for at least 12 h in advance. All participants signed an informed written consent form and gave their consent to the processing of personal data using the Lookin’Body 3.0 software.

Body height (H) was measured using professional electronical scales Tanita WB-300 in a standing position, while shoulders were in normal alignment and the head was in the horizontal Frankfurt plane. The participant stood upright on a horizontal surface, barefoot, with palms turned inwards and fingers pointing downwards. The height was measured from the sole of the feet to the top of the head. Body weight (W) was measured in light clothing using an electronic scale with an accuracy of 0.1 kg. Waist circumference at the umbilical level (WC) and hip circumference at the maximum level were determined using a measuring device with a stretched tape with an accuracy of 0.1 cm without any pressure on the body surface. Body mass index (BMI) was calculated as weight (kg) divided by square of the height (m^2^). Waist-to-hip ratio (WHR) and waist-to-height ratio (WHtR) were calculated as waist circumference (cm) divided by hip circumference (cm) and height (cm), respectively [28,29,30,31].

A body shape index (ABSI) was defined by WC/(BMI^2/3^ × height^1/2^). ABSI z-score was calculated based on the formula: ABSI z-score = (ABSI − ABSI_mean_)/ABSI_SD_ [13].

To assess the body composition, the following parameters were measured directly by bioimpedance analysis: fat free mass (FFM, kg and %); fat mass (FM, kg and %); visceral fat area (VFA, cm^2^); skeletal muscle mass (SMM, kg and %); extra-/intra- cellular and total body water (ICW, ECW, TBW, l and %); and chest circumference (CHC, cm); right arm circumference (RAC, cm); left arm circumference (LAC, cm); right leg circumference (RLC, cm); left leg circumference (LLC, cm) and arm muscle circumference (AMC, cm). 

Fat mass (kg) and fat free mass (kg) were taken to calculate fat mass index (kg/m^2^) and fat free mass index (kg/m^2^) as fat mass (kg) divided by square of the height (m^2^) or fat free mass (kg) divided by square of the height (m^2^).

### 2.3. Criteria for Obesity/Sarcopenia Diagnosis and Premature Mortality Risk

According to BMI, obesity was defined as BMI ≥ 30 kg/m^2^, underweight as BMI < 18.5 kg/m^2^, healthy weight in the value range of 18.5 and 25 and overweight in the range of 25 and 30 kg/m^2^ [32].

Fat mass/fat free mass ratio greater than 0.8 was used to define sarcopenic obesity. Values lower than 0.4 expressed metabolic health, values between 0.4 and 0.8 we considered as obesity [24,33].

Based on the FMI, fat deficit was defined by values ≤ 3.9 kg/m^2^, normal fat in the range of 3.9–8.2 kg/m^2^, excess fat in the range of 8.2–11.8 kg/m^2^, and high fat mass was defined by values ≥ 11.8 kg/m^2^. Based on the FFMI we defined skinny women with values ≤ 15 kg/m^2^, average women with FFMI between 14–17 kg/m^2^ (concurrently body fat in the range of 22–35%), fat women with FFMI between 15–18 kg/m^2^ (concurrently body fat in the range of 30–45%), athletic women with FFMI between 16–17 kg/m^2^ (concurrently body fat in the range of 18–25%), and bodybuilders with FFMI ≥ 18 kg/m^2^ [34,35].

According to Gonzales, and based on the FMI and FFMI, we determined sarcopenia (lower fat free mass), normal weight, obesity (high fat mass) and sarcopenic obesity (lower fat free mass and high fat mass) [35]. 

### 2.4. Statistical Analysis

Microsoft Office Excel 2016 (Los Angeles, CA, USA) in combination with XLSTAT (Version 2019) were used to process data. Statistical analysis was carried out using the STATISTICA 13 computer software (TIBCO Software, Inc., Palo Alto, CA, USA) and MedCalc ver. 20.104 software. Power analysis was conducted in G*Power to determine a sufficient sample size using an α value of 0.05, a power of 0.80, and a large effect size. The normality of variable distribution was checked with Shapiro–Wilk test. A descriptive analysis was carried out using the mean ± standard deviation. To evaluate the relationship between variables we used Pearson r correlation and linear regression equations. Levels of statistical significance were determined at *p* < 0.05. With a one-factor variance analysis (ANOVA), we tested the differences between variables and compared using Fisher’s Post Hoc Test. The risk of premature death (ABSI z-score) was calculated according to the methodology of Krakauer and Krakauer [20]. Premature mortality risk was classified into five categories: very low (<−0.868); low (between −0.868 and −0.272); average (between −0.272 and 0.229); high (between 0.229 and 0.798); and very high (>0.798) [20].

## 3. Results

A group of 303 young adult female volunteers participated in the study with a mean age of 21.73 ± 2.10 years. Among the anthropometric measures included, the mean BMI was 22.21 ± 3.38 kg/m^2^, WHR and WHtR 0.86 ± 0.05 and 0.48 ± 0.05, respectively. For fat-free mass, the mean was 44.82 ± 5.05 kg and the percentage of FFM was 72.77 ± 6.86%. Skeletal muscle mass determined by BIA was 24.52 ± 3.02 kg (percentage of SMM was 42.15 ± 4.75%). Among the parameters determining the amount and proportion of fat in the body, the mean fat mass was 17.45 ± 7.30 kg, the percentage of fat mass 27.23 ± 6.86%. For visceral fat area and waist circumference, the mean value was 69.98 ± 26.06 cm^2^ and 80.33 ± 9.09 cm, respectively. Characteristics of the participants are summarized in Table 1. 

### 3.1. Correlations between Indices Determining Obesity with Anthropometric Parameters and Premature Mortality Risk

Results of the Pearson’s correlation analyses of indices determining obesity with anthropometric parameters and premature mortality risk are shown in Table 2. For BMI, we found strong positive correlation with WC (0.907, *p* < 0.0001), HC (0.965, *p* < 0.0001), CHC (0.930, *p* < 0.0001), RAC (0.972, *p* < 0.0001), LAC (0.971, *p* < 0.0001), RLC (0.939, *p* < 0.0001), LLC (0.941, *p* < 0.0001), AMC (0.859, *p* < 0.0001), WHtR (0.948, *p* < 0.0001), VFA (0.870, *p* < 0.0001), and FM in kg (0.914, *p* < 0.0001). We also recorded a positive, but a moderate correlation with BMR (0.560, *p* < 0.0001), WHR (0.689, *p* < 0.0001), FFM in kg (0.560, *p* < 0.0001), as well as with percentage of FM (0.777, *p* < 0.0001) and SMM (0.559, *p* < 0.0001 in kg and 0.556, *p* < 0.0001 in %). BMI was in a moderate negative correlation with percentage of FFM (−0.777, *p* < 0.0001) and TBW (−0.776, *p* < 0.0001). Similar correlations were found for FMI with the exception of AMC, we found a moderate correlation (0.646, *p* < 0.0001). In addition, a strong negative correlation was demonstrated for percentage of FFM and TBW (−0.946, *p* < 0.0001 and −0.945, *p* < 0.0001, respectively). For FFMI we found strong positive correlations with AMC (0.893, *p* < 0.0001) and SMM in kg (0.800, *p* < 0.0001). Positive, but moderate correlations were identified in the case of WC (0.525, *p* < 0.0001), HC (0.706, *p* < 0.0001), CHC (0.772, *p* < 0.0001), RAC (0.726, *p* < 0.0001), LAC (0.730, *p* < 0.0001), RLC (0.683, *p* < 0.0001), LLC (0.688, *p* < 0.0001), WHtR (0.541, *p* < 0.0001), FFM in kg (0.785, *p* < 0.0001), FM in kg (0.415, *p* < 0.0001), and percentage of SMM (0.788, *p* < 0.0001). Correlation analyses of FM/FFM indicated a strong positive correlation only with WC (0.840, *p* < 0.0001), WHtR (0.891, *p* < 0.0001), VFA (0.928, *p* < 0.0001), and FM in kg (0.943, *p* < 0.0001) and percentage of FM (0.989, *p* < 0.0001). Strong, but negative correlations were recorded in the case of the percentage of FFM (−0.989, *p* < 0.0001) and TBW (−0.988, *p* < 0.0001). We did not confirm a strong correlation with ABSI or the risk of premature death for any of the indices. We found a positive moderate correlation only for FM/FFM ratio (ABSI and ABSI z-score, in both cases, r = 0.404, *p* < 0.0001) and a negative, but weak correlation for FFMI (ABSI and ABSI z-score, r = −0.367, *p* < 0.0001).

### 3.2. Assessment of Anthropometric Parameters and Indices Based on the Distribution of Participants According to Defined Cut-Offs of Variables Determining Obesity

Relationship between FMI, FFMI, FM/FFM, FMI and FFMI and BMI with anthropometric parameters and premature mortality risk is shown in Table 3.

As expected, a significant difference between the high fat mass group and the other groups according to FMI was confirmed in all parameters with the exception of height, FM (in kg), percentage of ICW and ECW. Similar significant results were also observed in the case of excess fat group, but without statistical evidence with the other groups in the case of BMR, height, FFM (in kg) as well as FM (in kg), likewise without evidence in the case of SMM. In addition, we did not observe a significant difference between the excess fat and high fat mass groups in relation to ABSI and the risk of premature death (*p* > 0.05).

According to FFMI, unequivocal and significant differences with the other categories were shown in the bodybuilder group with the highest diagnosed values of the monitored parameters, with the exception of the proportion of ICW and ECW from body water. We also revealed significant differences in the case of the fat group, but not in relation to the bodybuilder group in the case of WC, WHR, WHtR, VFA, FM and FMI. The risk of premature death was the highest in fat and skinny group (*p* < 0.05).

Similar to the previous case, most of the significant differences were found in the groups with a higher proportion of fat. According to the FM and FFM ratio, we found the highest and concurrently significant values in relation to the other groups in the sarcopenic obesity group (except percentage of FFM, ICW and ECW). However, significant differences were also shown in the obese group, but not in relation to the metabolically healthy group in the case of FFM (kg), FFMI, and SMM. The highest risk of premature death was confirmed in the obese group.

Based on the distribution of participants according to FMI and FFMI, the obese group had significantly higher values of the observed parameters in relation to the normal and sarcopenic groups, with the exception of the percentage of ICW and ECW. However, the risk of premature death was equally high in both the obese and sarcopenic groups, although the sarcopenic group showed, significantly, the lowest values of the observed parameters in most cases.

The distribution of participants according to BMI showed that the highest values of the observed parameters (with the exception of percentage of FFM, ECW and TBW) were in the obese group, and in most cases there were significant differences between the groups. Similar significant differences between groups were also observed in the overweight group. In the case of BMI distribution, it was confirmed that the underweight and obese groups have the highest risk of premature death (0.6816 and 0.8181, respectively).

### 3.3. Assessment of Risk of Premature Mortality Based on the Distribution of Participants according to Defined Cut-Offs of Variables Determining Obesity

Table 4 summarizes the results focused on the relationship between risk of premature death in young women and anthropometric variables. A very high risk of premature death was associated with the highest values in most of the assessed variables. We found significant differences in the very high-risk group compared to the other risk groups. For the young female group with the highest risk of premature mortality, mean weight was 66.31 kg (*p* < 0.05), WC 86.4 cm (*p* < 0.01), HC 94.9 cm (*p* < 0.05), WHR and WHtR 0.908 (*p* < 0.001) and 0.506 (*p* < 0.05), respectively. Mean VFA was 86.89 cm^2^ (*p* < 0.01), FM 21.12 kg (*p* < 0.05) and 30.9% (*p* < 0.001), FMI 7.24 kg/m^2^ (*p* < 0.01) and FM/FFM ratio 0.461 (*p* < 0.001). Among young women, the lowest values of FFM (69.1%, *p* < 0.001) a FFMI (15.45 kg/m^2^, *p* < 0.01), as well as TBW (50.57%, *p* < 0.05), were associated with the highest risk of premature mortality. The average risk of premature mortality was mostly associated with the lowest values of the monitored parameters, but without statistical significance. Moreover, in the group with a very low risk of premature mortality, the lowest mean values of WC, WHR, WHtR, VFA, FM, FMI, FM/FFM and ECW, and the highest values of BMR, AMC, FFM, FFMI, SMM, ICW and TBW were found. The differences were significant compared to the group of young women with a very high risk of premature death.

### 3.4. Assessment of the Obese Young Women Group According to Different Anthropometric Parameters and Indices Based on Adjusted Variables

We evaluated the group of obese young women based on selected adjusted variables (BMI, WHR, FM/FFM, FMI, FFMI, WC, VFA, FM in %). The variability of the values of the individual parameters is shown in Appendix A. The mean values of the parameters in women included in the obese groups based on adjusted variables are shown in Table 5. BMI showed, with the exception of FFM (%), significantly the highest mean values within the groups of obese women. Women classified as obese according to the FM/FFM ratio showed significantly the lowest mean values in most cases, with the exception of FFM (%). The results showed that the assessment of the presence of obesity and excess fat in young women according to various variables represents significant differences in the values of the monitored parameters.

## 4. Discussion

Based on the results and the values of the individual parameters, the group of young women was evaluated as a relatively high risk of premature mortality group (ABSI z-score 0.3120 ± 0.7122) at the current limit values of WC, WHR, WHtR (abdominal obesity), FM (%), FFM (%), FM/FFM (general/sarcopenic obesity) and optimal values of BMI, VFA, FMI and FFMI.

When diagnosing obesity, BMI is still recommended and used in practice, although it is known this may underestimate or overestimate the real state. Therefore, there is increasing discussion about modifying BMI as a diagnostic method or replacing it with other anthropometric indices and variables. As part of the correlation analysis, we found that the strongest positive association with FM (%) had the FM/FFM ratio followed by FMI, and only subsequently was shown correlation with BMI. In the case of FM (kg), a similar correlation was shown, but the strongest was for FMI, followed by the FM/FFM ratio and BMI. In the case of visceral fat, we found the same correlational relationships, the strongest with FMI, followed by FM/FFM and BMI. Our results suggest that FMI and FM/FFM indices correlate excellently with fat mass and visceral fat, better than BMI. Next, we found that WC correlated most strongly with FMI, then with BMI and finally with FM/FFM. We found the same sequence in the strength of the correlation in the case of WHtR. In the case of WHR, the strongest correlation was confirmed with FMI, then with FM/FFM and finally with BMI. Even in the case of abdominal obesity, FMI appears to be a relatively strong correlation, and stronger than BMI. In the case of fat-free mass and muscle mass, we found that the strongest correlations were shown by FFMI followed by BMI, FMI and FM/FFM (significantly only in the case of correlation with FFM). In the question of the risk of premature mortality, BMI has been shown to be in non-significant correlation. We found a moderate positive association in the case of FM/FFM and FMI, and a negative one with FFMI.

Stratification of women by BMI showed an increasing trend in the values of most variables with increasing BMI values, with the exception of %FFM (decreasing trend) and ABSI z-score (U-shaped curve). FMI, similar to BMI, showed a trend of increasing values in relation to the other monitored parameters, except for FFM (kg) and SSM (kg, %), which showed a J-shaped dependence and %FFM with a decreasing trend. 

In our study, as expected, we found that the highest FFMI values were in the group of women characterized as bodybuilders, but surprisingly, the athlete and fat groups had almost identical FFMI values. In the case of fatty individuals, this seems to be related to anabolic activity due to the body weight load on the musculoskeletal system. On average, obese adults have a larger mass of skeletal muscle than individuals of normal weight. The risk of sarcopenia was highest in the fat group adjusted according to FFMI. Furthermore, we found that significantly the highest BMI, WC, WHR, WHtR, VFA and FM values had fat and bodybuilder groups. At the same time, we must emphasize that it was the bodybuilder group that had the highest BMI values, not the fat group. Indicators of muscle mass (SMM, SMMI, FFM-kg) were highest in groups with the following order: bodybuilder; athlete; and fat. However, the percentage of FFM was highest in the athlete group followed by average and skinny groups. It clearly shows that parameters expressing quantitative and relative representation must be evaluated individually. The risk of premature mortality was highest in skinny and fat groups, which in turn suggests that high values of muscle mass in the simultaneous presence of low amounts of fat are not favorable for metabolic health, but also not vice versa.

The FM/FFM sarcopenic obesity indicator confirmed the increasing trend in values of individual indicator to the detriment of the sarcopenic obesity group. In the case of FM/FFM, with the exception of %FFM (decreasing trend) and FFMI (J-shaped curve), the values of the monitored parameters increased linearly in the sequence of metabolic health–obese–sarcopenic obesity.

The sarcopenia indicator through the FMI and FFMI relationship showed a decreasing trend in the values of the monitored parameters in the obesity-normal/health–sarcopenic group sequence, with the exception of the ABSI z-score (U-shaped dependence) and %FFM (bell-shaped dependence).

From the point of view of the risk of premature mortality, it was shown that for most variables, the risk of premature death increased with increasing values. We found a decreasing trend only in the case of %FFM and SMMI. In the case of FFM (kg) and SMM (kg, %), we found a U-shaped dependence.

When evaluating groups of obese women by different parameters, we found significant differences. BMI, as a universal and generally accepted diagnostic indicator of obesity, had the highest values of all monitored variables, with the exception of %FFM. FM/FFM on the contrary, as in its case we found, again with the exception of %FFM, the lowest values of variables. We found the most consistent values in the groups of women evaluated as obese according to the adjusted variables FMI, FFMI and WC, and also partly VFA, which may be related to abdominal obesity.

Conventional anthropometric methods have many limitations, and more accurate tools for estimating body fat tissue are still being sought. For the correct estimation of body composition, it is recommended to use a wide range of anthropometric indicators and indices. Body mass index is considered an unreliable indicator of fatness. An increase in BMI can be attributed to an increase in fat mass, as well as fat-free mass [36,37]. It should be emphasized that the relationship between BMI and disease risk varies between individuals as well as populations. BMI values for defining overweight and obesity are high for Asian populations, resulting in underdiagnosis of excess body fat in these populations [38]. Muscular individuals are classified as overweight due to their higher BMI, even if this is unjustified in terms of body fat. Furthermore, very short people can have high BMI values, which in no way may reflect overweight or obesity. In addition, susceptibility to disease risk factors also varies among individuals of different weights. In some individuals, overweight or obesity is associated with multiple risk factors, while in other individuals with more severe obesity, the occurrence of risk factors is limited. Older individuals tend to have a higher %body fat at a given BMI, which is why established BMI cut-offs may also be less accurate in older individuals (≥65 years) [39].

Several studies have found that high BMI is associated with an increased risk of chronic non-communicable diseases and all-cause mortality [40,41]. Morbidity risk by BMI forms a U- or J-shaped curve—low or high BMI increased this risk compared with near-middle BMI values. Krakauer and Krakauer [13] found that both low and high BMI values increased the risk of mortality in contrast to medium BMI values. They confirmed a U-shaped curve for the risk of death in relation to BMI and waist circumference. Our results also confirm the U-shaped curve for these two parameters. Even though BMI is the recommended indicator of obesity, the study by Zeng et al. [42] showed that it is a worse predictor of cardiovascular events than body fat percentage. Jabłonowska-Lietz et al. [43] reported only moderate correlations between body fat percentage and BMI. Our results also confirm these findings. Many factors lead to significant errors in the interpretation of BMI, including gender, race, greater muscle mass, changes in hydration status (especially extracellular fluid retention) [26]. There are many metabolically obese subjects of normal weight who have normal BMI but higher visceral adiposity as well as insulin resistance and increased cardiometabolic risk. As a result of muscle anabolism due to the higher load caused by higher body weight (obesity), obese adults have more skeletal muscle mass than lean individuals of the same age and sex. There is a direct correlation between BMI and muscle mass. Some obese individuals may experience sarcopenic obesity, which is a progressive loss of skeletal muscle. In the clinical setting, sarcopenic obesity is defined by higher fat mass compared to fat-free mass [24,44,45]. A higher FM is associated with a higher risk of premature death, which is also confirmed by our findings, on the contrary, a higher muscle mass reduces this risk [46].

Waist circumference indicates central (abdominal) obesity, but is largely dependent on body size. Together with WHR, they have been shown to have a similar association with incident diabetes as BMI [47] and together with WHtR have been shown to better discriminate cardiovascular disease risk [48] than BMI. According to the results of various studies, waist circumference was strongly correlated with BMI [49,50]. This was also confirmed in our study, when waist circumference increased linearly with BMI values. Amato, Guarnotta, and Giordano [26] demonstrated that waist circumference is strongly associated with visceral fat and abdominal adiposity, more so than BMI and WHR. This association was also confirmed in our case. Furthermore, Sato et al. [51] confirmed a strong correlation between BMI and WC and body weight, but not height. As reported by Krakauer and Krakauer [13], waist circumference is highly positively correlated with all parameters of body weight and body composition. Our findings support this claim. Waist circumference was positively associated with BMI, WHR, WHtR, VFA, FM (kg), SMM, as well as FFM (kg), but not percentage of FFM. We also found a positive association with FFMI, FMI, FM/FFM ratio and AMC. Thus, waist circumference is an index of overall body size and weight.

WHR was the first anthropometric index expressing the relationship between the distribution of fat in the body and the risk of various metabolic diseases. This index is predictive in relation to the risk of heart disease and diabetes. Its high values increase the risk of premature death, even though subjects have normal values of body mass index. The presence of excess fat in the abdominal area, disproportionate to total body fat, is an independent predictor of risk factors and morbidity. De Koning et al. [52] confirmed in their study that waist circumference and the ratio of waist circumference to hip circumference are statistically significantly associated with the risk of cardiovascular events. Yan et al. [53] emphasize that waist circumference and the waist-to-hip circumference are more important for assessing abdominal obesity compared to BMI. Most studies have pointed out the limited usefulness of the WHR index, but in practice its use in various clinical situations is still recommended [54,55].

Waist-to-height ratio determines the distribution of body fat, while its higher values indicate a higher risk of obesity-related disorders [56]. Studies suggest that waist-to-height ratio is a better predictor of visceral fat accumulation. The use of WHtR in public health screening is also recommended by Ashwell et al. [57], in whose study WHtR was a better predictor of mortality than other anthropometric indices.

Early research indicated a stronger relationship between ABSI and morbidity and early mortality from non-communicable diseases than BMI and WC [20]. The combination of BMI and WC is superior to BMI or waist circumference alone in assessing body composition and obesity as well as morbidity risk [58,59]. Among various anthropometric indicators, according to Krakauer and Krakauer [13], ABSI was the only one that was positively associated with fat mass and negatively with fat-free mass, while BMI and waist circumference were only positively associated with these parameters. Our findings confirm this. At the same time, in our study, when evaluating ABSI in relation to BMI, a U-shaped curve was confirmed, and in the case of waist circumference, a J-shaped curve. The advantage of ABSI is that it combines waist circumference, height and weight data. High ABSI values are indicative of higher waist circumference values and correspond to a more abdominal fat [13]. We reached similar results. A high ABSI value means that the WC is higher than expected for a given weight and height, while increasing body volume through more central fat accumulation.

A higher ABSI may indicate a larger VFA and a smaller proportion of muscle mass. In the study by Biolo et al. [60], a sufficient correlation between ABSI and BMI in the general population was not observed in individuals with overweight, or obesity regardless of gender. In addition, body shape index was negatively associated with fat-free mass in both men and women. Women and men with lower ABSI showed significantly greater FFMI than groups with higher ABSI for comparable BMI values. These findings support the hypothesis that abdominal fat deposition is associated with loss of skeletal muscle mass [61]. ABSI may not only be a marker of visceral obesity, but may also represent an index of reduced muscle mass. The authors of the study Biolo et al. [60] therefore hypothesize the usefulness of ABSI in the diagnosis of first degree sarcopenic obesity.

The association of ABSI with fat differs by sex. In contrast to men with an inverse association between ABSI and FM, in women this association was positive. This finding is partially confirmed by our results. This difference may be due to the different distribution of fat in the female and male body. While men are characterized by central abdominal obesity with a higher waist circumference, in women it is gluteal-femoral fat/obesity with a possible lower waist circumference at a given fat mass value [62]. Dhana et al. [63] supported this theory, as men had higher WC than women and consequently higher ABSI, while their BMI values were comparable to women. Women have more adipose tissue in the gynoid part of the body compared to men. This tissue is associated with lower metabolic risks [64,65].

Clear standards are needed for the diagnosis of obesity, which is also confirmed by O’Neill et al. [66] who found that the absence of a standard for defining obesity leads to various false negatives and false positives, and that almost 45% of Caucasian women are misclassified as non-obese. According to the authors, it is necessary to choose appropriate indicators for clinical use, depending on which element of the metabolic disease we are monitoring. Caitano Fontela et al. [67] concluded in their study that the use of anthropometric parameters cannot be an independent predictor of the underlying disease. Therefore, anthropometric indicators should be correlated with biochemical and clinical findings to accurately estimate risks for any metabolic disease. 

Our study has some limitations. The major limitation is the small number of participants, which could decrease the statistical power. The application of our results may be limited only to Caucasian ethnicity and young women. Further studies with larger samples should be conducted, so as to also include the confounding effects caused by potential covariates such as age, menstrual and hormonal status and physical activity. To our knowledge, this is the first study among Slovak young females evaluating the ability of anthropometric measurements to assess obesity and its type based on fat mass index and fat-free mass index, as well as the risk of premature mortality. The highly standardized anthropometric measurement procedures were major strengths of this study.

## 5. Conclusions

All anthropometric indices and indicators, whether conventional or emerging, rely on patient-specific characteristics such as fat and visceral fat mass or muscle mass. Our results suggest that FMI and FM/FFM indices correlate excellently with fat mass and visceral adipose tissue, better than BMI. Even in the case of abdominal obesity, FMI appears to have a relatively strong correlation, which is stronger than BMI. Stratification of women by BMI showed an increasing trend in the values of most variables with increasing BMI values, with the exception of %FFM (decreasing trend) and ABSI z-score (U-shaped curve). Our results confirm that parameters expressing quantitative (kg) and relative (%) representation must be evaluated individually. From the point of view of the risk of premature mortality, it was shown that for most variables, the risk of premature death increased with increasing values. BMI has been shown to be in non-significant correlation. We only found a decreasing trend in the case of %FFM and SMMI. In the case of FFM (kg) and SMM (kg, %) we found a U-shaped dependence.

In conclusion, we would like to emphasize that there is no agreement between commonly used indicators of total and abdominal obesity in its diagnosis and stratification of people into individual risk categories. In practice, BMI is still mainly used to assess body weight and diagnose obesity, but it has many discriminatory and confusing properties and is increasingly questioned. The waist circumference will also be evaluated in terms of the presence of abdominal obesity, but unfortunately other, easily defined anthropometric parameters and indices are not used. Currently, the possibilities of using the BIA method in the assessment of the body composition of adults are already commonly available, mainly due to affordability, therefore it would be advisable to introduce these examinations into screening, preventive and clinical practice for the purpose of a comprehensive evaluation of the health condition and body composition in terms of the proportionality of fat, but also muscle mass, which is often overlooked, even though it turns out that the negative impact on health and survival is associated not only with an excessive amount of adipose tissue, but also with lower muscle mass. Therefore, measuring the amount and proportion of fat and muscle tissue should become a generally accepted indicator for effective diagnosis and screening of obesity (sarcopenic obesity). 

## Figures and Tables

**Table 1 ijerph-19-12449-t001:** Characteristics of the participants (*n* = 303).

Variables	Mean	SD	95% CI	Min	Max
Age, years	21.73	2.10	21.50–21.97	18.00	25.00
BMR, kcal	1338	109	1325–1350	1019	1704
Height, m	1.674	0.06	1.67–1.68	1.51	1.89
Weight, kg	62.27	10.48	61.08–63.45	42.80	116.00
Waist Circumference, cm	80.33	9.09	79.30–81.36	65.60	119.30
Hip Circumference, cm	93.73	5.77	93.08–94.38	83.00	122.20
Chest Circumference, cm	88.57	6.01	87.89–89.25	76.20	115.30
Right Arm Circumference, cm	28.69	2.69	28.38–28.99	23.50	39.70
Left Arm Circumference, cm	28.55	2.69	28.24–28.85	23.50	39.10
Right Leg Circumference, cm	50.77	4.02	50.32–51.23	43.50	69.80
Left Leg Circumference, cm	50.96	4.14	50.49–51.43	43.40	70.70
Arm Muscle Circumference, cm	22.75	1.57	22.57–22.93	19.20	27.93
Body Mass Index, kg/m^2^	22.21	3.38	21.83–22.60	16.80	38.76
Waist-to-Hip Ratio	0.86	0.05	0.85–0.86	0.76	1.05
Waist-to-Height Ratio	0.48	0.05	0.47–0.49	0.40	0.69
Fat-free Mass, kg	44.82	5.05	44.25–45.40	30.10	61.80
Fat-free Mass, %	72.77	6.86	71.00–73.55	49.41	88.10
Fat Free Mass Index, kg/m^2^	15.99	1.41	15.83–16.14	12.81	20.65
Visceral Fat Area, cm^2^	69.98	26.06	67.04–72.93	14.34	171.05
Fat Mass, kg	17.45	7.30	16.62–18.27	6.40	54.20
Fat Mass, %	27.23	6.86	26.45–28.01	11.97	50.59
Fat Mass Index, kg/m^2^	6.23	2.55	5.94–6.52	2.33	18.11
Fat Mass/Fat Free Mass Ratio	0.39	0.14	0.37–0.40	0.14	1.02
Skeletal Muscle Mass, kg	24.52	3.02	24.18–24.87	15.63	34.32
Skeletal Muscle Mass, %	42.15	4.75	41.61–42.69	28.20	58.00
Extracellular Water, L	12.48	1.39	12.33–12.64	8.50	17.30
ECW/TBW, %	38.04	0.49	37.99–38.10	36.13	39.23
Intracellular Water, L	20.34	2.31	20.08–20.60	13.50	27.90
ICW/TBW, %	61.96	0.49	61.90–62.01	60.77	63.87
Total Body Water, L	32.82	3.69	32.40–33.24	22.00	45.20
TBW/W, %	53.29	5.08	52.72–53.87	36.19	64.41
ABSI, m^11/6^ kg^−2/3^	0.0788	0.0030	0.078–0.079	0.0707	0.0874
ABSI z-score	0.3120	0.7122	0.23–0.39	−1.6170	2.3670

ECW, extracellular water; ICW, intracellular water; TBW, total body water.

**Table 2 ijerph-19-12449-t002:** Correlation between indices determining obesity and anthropometric parameters.

Variables	Body Mass Index (kg/m^2^)		Fat Mass Index (kg/m^2^)		Fat Free Mass Index (kg/m^2^)		Fat Mass/Fat Free Mass Ratio	
	r	*p*	r	*p*	r	*p*	r	*p*
ABSI, m^11/6^ kg^−2/3^	0.073	0.2077	0.300	<0.0001	−0.367	<0.0001	0.404	<0.0001
ABSI z-score	0.073	0.2066	0.300	<0.0001	−0.367	<0.0001	0.404	<0.0001
BMR, kcal	0.560	<0.0001	0.309	<0.0001	0.784	<0.0001	0.119	0.0378
Height, m	−0.025	0.6621	−0.036	0.5271	0.006	0.924	−0.059	0.3035
Weight, kg	0.907	<0.0001	0.834	<0.0001	0.667	<0.0001	0.715	<0.0001
WC, cm	0.907	<0.0001	0.912	<0.0001	0.525	<0.0001	0.840	<0.0001
HC, cm	0.965	<0.0001	0.890	<0.0001	0.706	<0.0001	0.770	<0.0001
CHC, cm	0.930	<0.0001	0.807	<0.0001	0.772	<0.0001	0.665	<0.0001
RAC, cm	0.972	<0.0001	0.888	<0.0001	0.726	<0.0001	0.770	<0.0001
LAC, cm	0.971	<0.0001	0.884	<0.0001	0.730	<0.0001	0.765	<0.0001
RLC, cm	0.939	<0.0001	0.868	<0.0001	0.683	<0.0001	0.753	<0.0001
LLC, cm	0.941	<0.0001	0.868	<0.0001	0.688	<0.0001	0.752	<0.0001
AMC, cm	0.859	<0.0001	0.646	<0.0001	0.893	<0.0001	0.467	<0.0001
Waist-to-Hip Ratio	0.689	<0.0001	0.781	<0.0001	0.240	<0.0001	0.778	<0.0001
Waist-to-Height Ratio	0.948	<0.0001	0.958	<0.0001	0.541	<0.0001	0.891	<0.0001
Fat-free Mass, kg	0.560	<0.0001	0.309	<0.0001	0.785	<0.0001	0.119	0.0378
Fat-free Mass, %	−0.777	<0.0001	−0.946	<0.0001	−0.154	0.0073	−0.989	<0.0001
Visceral Fat Area, cm^2^	0.870	<0.0001	0.952	<0.0001	0.365	<0.0001	0.928	<0.0001
Fat Mass, kg	0.914	<0.0001	0.983	<0.0001	0.415	<0.0001	0.943	<0.0001
Fat Mass, %	0.777	<0.0001	0.946	<0.0001	0.154	0.0073	0.989	<0.0001
Skeletal Muscle Mass, kg	0.559	<0.0001	0.299	<0.0001	0.800	<0.0001	0.105	0.0678
Skeletal Muscle Mass, %	0.556	<0.0001	0.301	<0.0001	0.788	<0.0001	0.110	0.0564
ICW/TBW, %	0.168	0.0034	0.031	0.5915	0.347	<0.0001	−0.045	0.4333
ECW/TBW, %	−0.168	0.0034	−0.031	0.5915	−0.347	<0.0001	0.045	0.4333
TBW/W, %	−0.776	<0.0001	−0.945	<0.0001	−0.153	0.0075	−0.988	<0.0001

ABSI, a body shape index; BMR, basal metabolic rate; WC, waist circumference; HC, hip circumference; CHC, chest circumference; RAC, right arm circumference; LAC, left arm circumference; RLC, right leg circumference; LLC, left leg circumference; AMC, arm muscle circumference; ICW, intracellular water; ECW, extracellular water; TBW, total body water.

**Table 3 ijerph-19-12449-t003:** Relationship between FMI, FFMI, FM/FFM, FMI and FFMI and Body Mass Index with anthropometric parameters and premature mortality risk.

**Variables**	**FMI**				**FFMI**					
	**Fat Deficit**	**Normal**	**Excess Fat**	**High Fat Mass**	**Skinny**	**Average**	**Athlete**		**Fat**	**Body Builder**
BMR, kcal	1339 ^a^	1325 ^a^	1350 ^a^	1510 ^b^	1236 ^a^	1347 ^b^	1396 ^c^		1369 ^bc^	1525 ^d^
Height, m	1.69 ^a^	1.67	1.66 ^b^	1.69	1.67	1.68	1.68		1.66	1.67
Weight, kg	54.30 ^a^	60.31 ^b^	71.50 ^c^	92.70 ^d^	55.13 ^a^	61.20 ^b^	59.78 ^b^		75.19 ^c^	81.05 ^d^
WC, cm	71.60 ^a^	78.60 ^b^	90.41 ^c^	107.19 ^d^	75.68 ^a^	78.97 ^b^	75.41 ^ac^		93.45 ^d^	94.81 ^d^
HC, cm	88.63 ^a^	92.64 ^b^	99.66 ^c^	110.64 ^d^	89.58 ^a^	93.09 ^b^	92.13 ^b^		101.74 ^c^	104.39 ^d^
CHC, cm	84.45 ^a^	87.36 ^b^	94.16 ^c^	105.48 ^d^	83.50 ^a^	88.14 ^b^	87.84 ^b^		96.32 ^c^	100.37 ^d^
RAC, cm	26.19 ^a^	28.18 ^b^	31.72 ^c^	36.22 ^d^	26.56 ^a^	28.45 ^b^	27.88 ^b^		32.62 ^c^	33.70 ^d^
LAC, cm	26.09 ^a^	28.03 ^b^	31.58 ^c^	36.06 ^d^	26.41 ^a^	28.30 ^b^	27.82 ^b^		32.47 ^c^	33.58 ^d^
RLC, cm	47.15 ^a^	50.07 ^b^	54.84 ^c^	61.99 ^d^	47.98 ^a^	50.36 ^b^	49.63 ^b^		56.22 ^c^	57.82 ^d^
LLC, cm	47.19 ^a^	50.24 ^b^	55.11 ^c^	62.55 ^d^	48.05 ^a^	50.54 ^b^	49.80 ^b^		56.54 ^c^	58.24 ^d^
AMC, cm	22.00 ^a^	22.47 ^b^	23.97 ^c^	26.29 ^d^	21.15 ^a^	22.72 ^b^	23.17 ^c^		24.30 ^d^	26.10 ^e^
BMI, kg/m^2^	18.98 ^a^	21.55 ^b^	26.04 ^c^	32.30 ^d^	19.73 ^a^	21.78 ^b^	21.18 ^b^		27.25 ^c^	28.76 ^d^
WHR	0.81 ^a^	0.85 ^b^	0.91 ^c^	0.97 ^d^	0.84 ^a^	0.85 ^a^	0.82 ^b^		0.92 ^c^	0.90 ^c^
WHtR	0.42 ^a^	0.47 ^b^	0.55 ^c^	0.63 ^d^	0.45 ^a^	0.47 ^b^	0.45 ^ac^		0.56 ^d^	0.57 ^d^
FFM, kg	44.90 ^a^	44.22 ^a^	45.37 ^a^	52.78 ^b^	40.11 ^a^	45.23 ^b^	47.53 ^c^		46.28 ^bc^	53.50 ^d^
FFM, %	82.64 ^a^	73.48 ^b^	63.45 ^c^	57.01 ^d^	73.13 ^a^	74.23 ^a^	79.54 ^b^		61.98 ^c^	67.59 ^d^
FFMI, kg/m^2^	15.70 ^a^	15.80 ^a^	16.53 ^b^	18.37 ^c^	14.34 ^a^	16.09 ^b^	16.84 ^c^		16.78 ^c^	19.06 ^d^
VFA, cm^2^	38.44 ^a^	66.02 ^b^	101.94 ^c^	142.03 ^d^	62.23 ^a^	65.07 ^a^	50.56 ^b^		109.13 ^c^	103.12 ^c^
FM, kg	9.4	16.09	26.13	39.92	15.01 ^ac^	15.97 ^a^	12.25 ^c^		28.91 ^d^	27.54 ^d^
FM, %	17.37 ^a^	26.53 ^b^	36.55 ^c^	42.98 ^d^	26.87 ^a^	25.78 ^a^	20.47 ^b^		38.03 ^b^	32.41 ^b^
FMI, kg/m^2^	3.28 ^a^	5.75 ^b^	9.51 ^c^	13.93 ^d^	5.39 ^a^	5.69 ^a^	4.34 ^b^		10.47 ^c^	9.70 ^c^
FM/FFM	0.21 ^a^	0.37 ^b^	0.58 ^c^	0.76 ^d^	0.38 ^a^	0.35 ^a^	0.26 ^b^		0.62 ^c^	0.51 ^d^
SMM, kg	24.63 ^a^	24.16 ^a^	24.83 ^a^	29.21 ^b^	21.64 ^a^	24.79 ^b^	26.19 ^c^		25.37 ^bc^	29.79 ^d^
SMM, %	42.30 ^a^	41.58 ^a^	42.61 ^a^	49.56 ^b^	37.70 ^a^	42.54 ^b^	44.77 ^c^		43.46 ^bc^	50.36 ^d^
ICW/TBW, %	61.99	61.94	62.03	62.05	61.69 ^a^	62.02 ^b^	62.01 ^bc^		62.06 ^bc^	62.23 ^c^
ECW/TBW, %	38.01	38.06	37.98	37.95	38.31 ^a^	37.98 ^b^	38.00 ^bc^		37.94 ^bc^	37.77 ^c^
TBW/W, %	60.62 ^a^	53.81 ^b^	46.38 ^c^	41.68 ^d^	53.57 ^a^	54.35 ^a^	58.34 ^b^		45.29 ^c^	49.50 ^d^
ABSI m^11/6^ kg^−2/3^	0.0775 ^a^	0.0786 ^b^	0.0801 ^c^	0.0814 ^c^	0.0803 ^a^	0.0783 ^b^	0.0761 ^c^		0.0802 ^a^	0.0779 ^b^
ABSI z-score	0.0064 ^a^	0.2777 ^b^	0.6212 ^c^	0.9527 ^c^	0.6794 ^a^	0.1923 ^b^	−0.3268 ^c^		0.6540 ^a^	0.1109 ^b^
**Variables**	**FM/FFM**			**FMI and FFMI**			**Body Mass Index**			
	**Metabolic Health**	**Obese**	**Sarcopenic Obesity**	**Obesity**	**Normal**	**Sarcopenia**	**Underweight**	**Healthy Weight**	**Overweight**	**Obese**
BMR, kcal	1329 ^a^	1348 ^a^	1510 ^b^	1510 ^a^	1362 ^b^	1236 ^c^	1228 ^a^	1328 ^b^	1404 ^c^	1558 ^d^
Height, m	1.67	1.67	1.68	1.69	1.67	1.67	1.68	1.67	1.66	1.70
Weight, kg	57.66 ^a^	68.62 ^b^	103.10 ^c^	92.70 ^a^	62.78 ^b^	55.13 ^c^	50.33 ^a^	60.18 ^b^	74.15 ^c^	95.98 ^d^
WC, cm	75.42 ^a^	87.40 ^b^	111.83 ^c^	107.19 ^a^	80.20 ^b^	75.68 ^c^	70.98 ^a^	78.39 ^b^	91.21 ^c^	108.89 ^d^
HC, cm	91.07 ^a^	97.39 ^b^	117.63 ^c^	110.64 ^a^	94.08 ^b^	89.58 ^c^	86.49 ^a^	92.52 ^b^	101.20 ^c^	112.43 ^d^
CHC, cm	86.08 ^a^	92.02 ^b^	110.23 ^c^	105.48 ^a^	89.25 ^b^	83.50 ^c^	81.03 ^a^	87.42 ^b^	95.79 ^c^	107.31 ^d^
RAC, cm	27.40 ^a^	30.51 ^b^	38.17 ^c^	36.22 ^a^	28.94 ^b^	26.56 ^c^	25.08 ^a^	28.16 ^b^	32.27 ^c^	36.88 ^d^
LAC, cm	27.27 ^a^	30.35 ^b^	38.10 ^c^	36.06 ^a^	28.81 ^b^	26.41 ^c^	24.98 ^a^	28.02 ^b^	32.13 ^c^	36.71 ^d^
RLC, cm	48.99 ^a^	53.21 ^b^	67.20 ^c^	61.99 ^a^	51.02 ^b^	47.98 ^c^	45.73 ^a^	49.95 ^b^	55.97 ^c^	63.30 ^d^
LLC, cm	49.12 ^a^	53.47 ^b^	67.97 ^c^	62.55 ^a^	51.22 ^b^	48.05 ^c^	45.73 ^a^	50.12 ^b^	56.26 ^c^	63.92 ^d^
AMC, cm	22.27 ^a^	23.42 ^b^	26.71 ^c^	26.29 ^a^	23.08 ^b^	21.15 ^c^	20.65 ^a^	22.51 ^b^	24.56 ^c^	26.85 ^d^
BMI, kg/m^2^	20.60 ^a^	24.44 ^b^	36.43 ^c^	32.30 ^a^	22.42 ^b^	19.73 ^c^	17.85 ^a^	21.48 ^b^	26.85 ^c^	33.23 ^d^
WHR	0.83 ^a^	0.90 ^b^	0.95 ^c^	0.97 ^a^	0.85 ^b^	0.84 ^b^	0.82 ^a^	0.85 ^b^	0.90 ^c^	0.97 ^d^
WHtR	0.45 ^a^	0.52 ^b^	0.67 ^c^	0.63 ^a^	0.48 ^b^	0.45 ^c^	0.42 ^a^	0.47 ^b^	0.55 ^c^	0.64 ^d^
FFM, kg	44.41 ^a^	45.28 ^a^	52.83 ^b^	52.78 ^a^	45.95 ^b^	40.11 ^c^	39.76 ^a^	44.39 ^b^	47.91 ^c^	55.02 ^d^
FFM, %	77.13 ^a^	66.32 ^b^	51.13 ^c^	57.01 ^a^	73.68 ^b^	73.13 ^b^	78.95 ^a^	73.98 ^b^	64.72 ^c^	57.44 ^d^
FFMI, kg/m^2^	15.85 ^a^	16.13 ^a^	18.65 ^b^	18.37 ^a^	16.41 ^b^	14.34 ^c^	14.09 ^a^	15.84 ^b^	17.35 ^c^	19.02 ^d^
VFA, cm^2^	54.35 ^a^	92.93 ^b^	154.72 ^c^	142.03 ^a^	67.99 ^b^	62.23 ^c^	44.95 ^a^	64.56 ^b^	101.78 ^c^	144.39 ^d^
FM, kg	13.25 ^a^	23.34 ^b^	50.27 ^c^	39.92 ^a^	16.83 ^b^	15.01 ^c^	10.57 ^a^	15.79 ^b^	26.24 ^c^	40.97 ^d^
FM, %	22.87 ^a^	33.67 ^b^	48.89 ^c^	42.98 ^a^	26.32 ^b^	26.87 ^b^	21.06 ^a^	26.02 ^b^	35.28 ^c^	42.56 ^d^
FMI, kg/m^2^	4.74 ^a^	8.32 ^b^	17.78 ^c^	13.93 ^a^	6.02 ^b^	5.39 ^c^	3.76 ^a^	5.64 ^b^	9.50 ^c^	14.21 ^d^
FM/FFM	0.30 ^a^	0.51 ^b^	0.96 ^c^	0.76 ^a^	0.37 ^b^	0.38 ^b^	0.27 ^a^	0.36 ^b^	0.55 ^c^	0.75 ^d^
SMM, kg	24.30 ^a^	24.76 ^a^	29.09 ^b^	29.21 ^a^	25.23 ^b^	21.64 ^c^	21.47 ^a^	24.27 ^b^	26.36 ^c^	30.57 ^d^
SMM, %	41.79 ^a^	42.54 ^a^	49.57 ^b^	49.56 ^a^	43.22 ^b^	37.70 ^c^	37.42 ^a^	41.74 ^b^	45.03 ^c^	51.68 ^d^
ICW/TBW, %	61.98	61.93	61.79	62.05 ^a^	62.05 ^a^	61.69 ^b^	61.67 ^a^	61.97 ^b^	62.03 ^b^	62.12 ^b^
ECW/TBW, %	38.02	38.07	38.21	37.95 ^a^	37.96 ^a^	38.31 ^b^	38.33 ^a^	38.03 ^b^	37.97 ^b^	37.88 ^b^
TBW/W, %	56.52 ^a^	48.52 ^b^	37.39 ^c^	41.68 ^a^	53.96 ^b^	53.57 ^b^	57.94 ^a^	54.18 ^b^	47.35 ^c^	41.99 ^d^
ABSI, m^11/6^ kg^−2/3^	0.0778 ^a^	0.0804 ^b^	0.0784	0.0814 ^a^	0.0781 ^b^	0.0803 ^ac^	0.0804 ^a^	0.0785 ^b^	0.0790	0.0809 ^a^
ABSI z-score	0.0826 ^a^	0.6838 ^b^	0.2321	0.9527 ^a^	0.1417 ^b^	0.6794 ^ac^	0.6816 ^a^	0.2466 ^b^	0.3622	0.8181 ^a^

BMR, basal metabolic rate; WC, waist circumference; HC, hip circumference; CHC, chest circumference; RAC, right arm circumference; LAC, left arm circumference; RLC, right leg circumference; LLC, left leg circumference; AMC, arm muscle circumference; BMI, body mass index; WHR, waist-to-hip ratio; WHtR, waist-to-height ratio; FFM, fat-free mass; FFMI, fat free mass index; VFA, visceral fat area; FM, fat mass; FMI, fat mass index; SMM, skeletal muscle mass; ICW, intracellular water; ECW, extracellular water; TBW, total body water; ABSI, body shape index; ^a–d^, different symbols in the row mean significant differences.

**Table 4 ijerph-19-12449-t004:** Relationship between ABSI z-score with anthropometric parameters.

Variables	Premature Mortality Risk				
	Very Low	Low	Average	High	Very High
BMR, kcal	1405 ^a^	1358 ^ac^	1311 ^b^	1334 ^bc^	1346
Height, m	1.64 ^a^	1.65 ^a^	1.65 ^a^	1.69 ^b^	1.71 ^c^
Weight, kg	60.12 ^acd^	60.94 ^ac^	59.26 ^ab^	62.77 ^c^	66.31 ^d^
WC, cm	73.65 ^a^	76.86 ^a^	77.33 ^a^	81.07 ^b^	86.40 ^c^
HC, cm	93.50	93.87	92.53 ^a^	93.80	94.90 ^b^
CHC, cm	87.99	88.10 ^a^	86.96 ^a^	88.60 ^a^	90.79 ^b^
RAC, cm	28.44	28.59	28.07 ^a^	28.65 ^a^	29.54 ^b^
LAC, cm	28.26	28.50	27.91 ^a^	28.53 ^a^	29.36 ^b^
RLC, cm	51.07	51.18	50.03	50.85	51.13
LLC, cm	51.30	51.46	50.26	50.99	51.27
AMC, cm	23.38	23.02 ^a^	22.42 ^b^	22.63	22.97 ^a^
BMI, kg/m^2^	22.24	22.52	21.76	22.6	22.69
WHR	0.787 ^a^	0.818 ^b^	0.834 ^c^	0.862 ^d^	0.908 ^e^
WHtR	0.448 ^a^	0.467 ^ab^	0.469 ^ab^	0.481 ^b^	0.506 ^c^
FFM, kg	47.95 ^a^	45.75 ^ac^	43.58 ^b^	44.67 ^bc^	45.20
FFM, %	79.83 ^a^	75.46 ^b^	74.03 ^b^	72.09 ^c^	69.10 ^d^
FFMI, kg/m^2^	17.73 ^a^	16.88 ^b^	15.96 ^c^	15.68 ^cd^	15.45 ^d^
VFA, cm^2^	45.52 ^a^	59.86 ^ab^	62.51 ^b^	72.29 ^c^	86.89 ^d^
FM, kg	12.17 ^a^	15.19 ^a^	15.67 ^a^	18.10 ^b^	21.12 ^c^
FM, %	20.17 ^a^	24.55 ^b^	25.97 ^b^	27.92 ^c^	30.90 ^d^
FMI, kg/m^2^	4.51 ^a^	5.63 ^ab^	5.80 ^ab^	6.38 ^b^	7.24 ^c^
FM/FFM	0.257 ^a^	0.332 ^ab^	0.361 ^bc^	0.401 ^c^	0.461 ^d^
SMM, kg	26.55 ^a^	25.10 ^ac^	23.79 ^b^	24.43 ^bc^	24.71
SMM, %	45.20 ^a^	43.05 ^ac^	40.99 ^b^	41.00 ^bc^	42.46
ICW/TBW, %	62.33 ^a^	62.01 ^b^	61.91 ^b^	61.95 ^b^	61.92 ^b^
ECW/TBW, %	37.67 ^a^	37.99 ^b^	38.09 ^b^	38.05 ^b^	38.08 ^b^
TBW/W, %	58.49 ^a^	55.29 ^b^	54.24 ^b^	52.78 ^c^	50.57 ^d^
ABSI, m^11/6^ kg^−2/3^	0.0729 ^a^	0.0753 ^b^	0.0774 ^c^	0.0796 ^d^	0.0828 ^e^
ABSI z-score	−1.106	−0.5296	−0.0188	0.51865	1.2763

BMR, basal metabolic rate; WC, waist circumference; HC, hip circumference; CHC, chest circumference; RAC, right arm circumference; LAC, left arm circumference; RLC, right leg circumference; LLC, left leg circumference; AMC, arm muscle circumference; BMI, body mass index; WHR, waist-to-hip ratio; WHtR, waist-to-height ratio; FFM, fat-free mass; FFMI, fat free mass index; VFA, visceral fat area; FM, fat mass; FMI, fat mass index; SMM, skeletal muscle mass; ICW, intracellular water; ECW, extracellular water; TBW, total body water; ABSI, body shape index. ^a–e^, different symbols in the row mean significant differences.

**Table 5 ijerph-19-12449-t005:** Assessment of variables in groups of young women defined as obese according to different anthropometric parameters and indices (mean).

Variables/Adjusted Variables	BMI–a	WHR–b	FM/FFM–c	FMI–d	FFMI–e	WC–f	VFA–g	FM (%)–h
BMI, kg/m^2^	33.23 ^b–h^	26.52 ^acg^	24.44 ^a–h^	27.90 ^ach^	27.25 ^ac^	27.65 ^ach^	28.68 ^a–ch^	25.93 ^acdfg^
WHR	0.97 ^bc–fh^	0.94 ^aceh^	0.90 ^a–g^	0.93 ^acgh^	0.92 ^a–cg^	0.94 ^ach^	0.95 ^c–eh^	0.91 ^abdfg^
FM/FFM	0.75 ^b–h^	0.59 ^acg^	0.51 ^a–h^	0.63 ^ach^	0.62 ^ac^	0.60 ^acg^	0.66 ^a–cfh^	0.58 ^acdg^
FMI, kg/m^2^	14.21 ^b–h^	9.88 ^acg^	8.32 ^a–h^	10.83 ^ach^	10.47 ^ac^	10.40 ^ac^	11.43 ^a–ch^	9.54 ^acdg^
FFMI, kg/m^2^	19.02 ^b–h^	16.65 ^ac^	16.13 ^abdfg^	17.08 ^ach^	16.78 ^a^	17.26 ^ach^	17.25 ^ach^	16.39 ^adfg^
WC, cm	108.89 ^b–h^	95.05 ^acgh^	87.40 ^a–h^	95.41 ^acgh^	93.45 ^acg^	96.66 ^ach^	99.64 ^a–eh^	90.69 ^a–dfg^
VFA, cm^2^	144.39 ^b–h^	110.36 ^acg^	92.93 ^a–h^	113.88 ^acgh^	109.13 ^acg^	113.99 ^ach^	123.75 ^a–eh^	102.70 ^acdfg^
BMR, kcal	1558 ^b–h^	1407 ^ach^	1348 ^abdfg^	1398 ^ac^	1370 ^afg^	1446 ^aceh^	1451 ^aceh^	1358 ^abfg^
SMM, %	51.68 ^b–h^	45.11 ^ach^	42.54 ^abdfg^	44.68 ^ac^	43.46 ^afg^	46.80 ^aceh^	46.96 ^aceh^	42.95 ^abfg^
FFM, %	57.44 ^b–h^	63.54 ^acdg^	66.32 ^a–h^	61.53 ^a–ch^	61.98 ^ac^	62.99 ^acg^	60.61 ^a–cfh^	63.69 ^acdg^
AMC, cm	26.85 ^b–h^	24.44 ^ach^	23.42 ^a–g^	24.66 ^ach^	24.30 ^ac^	24.97 ^ach^	25.09 ^ach^	23.82 ^abdfg^
ABSI, m^11/6^ kg^−2/3^	0.0809 ^ns^	0.0825 ^c–eh^	0.0804 ^bfg^	0.0805 ^bg^	0.0802 ^bg^	0.0815 ^ch^	0.0819 ^c–eh^	0.0803 ^bfg^
ABSI z–score	0.8181 ^ns^	1.1954 ^c–eh^	0.6838 ^bfg^	0.7199 ^bg^	0.6540 ^bg^	0.9501 ^ch^	1.0534 ^c–eh^	0.6781 ^bfg^

BMI, body mass index; WHR, waist-to-hip ratio; FM/FFM; fat mass/fat-free mass ratio; FMI, fat mass index; FFMI, fat free mass index; WC, waist circumference; VFA, visceral fat area; BMR, basal metabolic rate; SMM, skeletal muscle mass; FFM, fat-free mass; AMC, arm muscle circumference; ABSI, body shape index; ns, non-significant; ^a–h^, symbols in the row mean significant differences between adjusted variables.

## Data Availability

Not applicable.

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
