# Peer review of "The Use of Different Anthropometric Indices to Assess the Body Composition of Young Women in Relation to the Incidence of Obesity, Sarcopenia and the Premature Mortality Risk"

_ijerph, 2022, doi:10.3390/ijerph191912449_

Round 1

Reviewer 1 Report

This study evaluated the stratification of young women based on the assessment of body composition according to several currently recommended anthropometric indices and parameters, as well as the presence of obesity, sarcopenic obesity and the risk of premature death. The topic is novel and deserves publication.

Reviewer 2 Report

The study is very extensive. I feel some parts could be omitted, but it was the authors' decision (and I respect it) to include so much data and analysis; of course, it is not a mistake, but in some way, the study is not easy to follow. However, it has a good substantive base with good references to the available literature observation. For me, the main issue of this paper is the lack (or too low) exposition of provided results of this study. When you refer to it, you did it in general, and the readers could have difficulty extracting the information from your manuscript.

Line 23 – "…several anthropometric parameters and indices…" be precise when You describe your results. Which one?

Line 29-30 – avoid one-sentence paragraphs.

Line 117 –"height" of what? Add Body height.

158-161 – I think in the section on statistical analysis (Line 162), you should describe the method used in statistics.

Line 183 - Table 1 – I do not feel median is necessary when You provide means for continuous data.

Line 252 – table 3. You need to improve table 3. The column names and the data slide apart. This makes the data difficult to read.

Line 280 – delete an additional dot in the table name.

Line 307 – Discussion – I think You should start with the paragraph which summarizes your results, but only the essential observations.

Line 454 – Generally, I agree with your statement in conclusion, but I think there is a lack of direct reference to your results. I think you need to emphasize it. You need to answer precisely what your results show.

Author Response

Dear Reviewer,

Thank you for revising our manuscript entitled “The Use of Different Anthropometric Indices to Assess the Body Composition of Young Women in Relation to the Incidence of Obesity, Sarcopenia and the Premature Mortality Risk”.

We greatly appreciate the time and efforts to review our manuscript and we agree that the proposed changes will contribute to the improvement of our manuscript. We have addressed all issues indicated in reviews, and we believe that the revised version can meet the journal publication requirements.

Please find our responses to the Reviewers comments attached. The changes made in the text are highlighted in red.

Yours Sincerely,

Martina Gažarová

Point 1: The study is very extensive. I feel some parts could be omitted, but it was the authors' decision (and I respect it) to include so much data and analysis; of course, it is not a mistake, but in some way, the study is not easy to follow. However, it has a good substantive base with good references to the available literature observation. For me, the main issue of this paper is the lack (or too low) exposition of provided results of this study. When you refer to it, you did it in general, and the readers could have difficulty extracting the information from your manuscript.

Response 1: We thank the reviewer for his/her statement, opinion and for all the valuable advice. We really appreciate it. We agree that our manuscript has a broad focus. We are also grateful that the reviewer respects our decision to present a large number of results and data. Regarding the interpretation of our results, we tried to improve this part (see discussion and conclusion).

Point 2:

Line 23 – "…several anthropometric parameters and indices…" be precise when You describe your results. Which one? Thank you for comment. It was corrected (page 1, lines 25-26).

Line 29-30 – avoid one-sentence paragraphs. Thank you for your suggestion. It was corrected.

Line 117 –"height" of what? Add Body height. Thank you for your suggestion. The word „Body” was added (page 3, line 135).

158-161 – I think in the section on statistical analysis (Line 162), you should describe the method used in statistics. Thank you for your suggestion. We introduced this part in the section Statistical analysis (page 4, lines 187-190).

Line 183 - Table 1 – I do not feel median is necessary when You provide means for continuous data. Thank you for your suggestion, we removed the column stating the median.

Line 252 – table 3. You need to improve table 3. The column names and the data slide apart. This makes the data difficult to read. Thank you for your suggestion, we adjusted all tables (their size), variable names, data to make them more readable.

Line 280 – delete an additional dot in the table name. Thank you, we corrected it (page 9, line 301).

Line 307 – Discussion – I think You should start with the paragraph which summarizes your results, but only the essential observations. Thank you for your suggestion, we tried to improve this part (page 11, lines 329-394).

Line 454 – Generally, I agree with your statement in conclusion, but I think there is a lack of direct reference to your results. I think you need to emphasize it. You need to answer precisely what your results show. Thank you for your suggestion, we tried to improve this part (page 14, lines 520-531).

Reviewer 3 Report

Abstract

Line 15 does not need to start off with a number.

Introduction

Lines 30 – 43 should be combined as one paragraph.

Sarcopenic obesity needs to be defined, and its significance needs to be elaborated on. This is the cornerstone of your study, and there is nothing about it in the introduction!

Lines 44 – 84 are not written like an introduction. Way too many short paragraphs and independent thoughts. What is the overall focus of the manuscript, and why is this study important?

Methods

In the Participants and Study Design section, what happened to the other 128 individuals who participated in the cross-sectional observational study? Also, is this a secondary analysis?

Stats

What is the number of participants presented by the power analysis?

In the stats section, there is no mention of a predictor model employed. This model is imperative when using independent variables to predict an outcome. Essentially, a regression model needs to be run.

All the tables need to be redone. They are crowded and cumbersome and are hard to follow/interpret. the variables and headings are bad and appear unprofessional and not readable. The symbols used for significance need to be something else besides abcd etc.

Also, the results section needs to be rewritten. The tables are what provide the bulk of the results. Therefore, the text highlights the main significant findings.

Discussion

Your main findings need to be presented first. Then each paragraph after that highlights each of your main findings.

There are too many directions that the discussion is flowing, and there is no focus.

The conclusion is too broad, and is hard for the reader to take away the actual significance of the paper. 

Author Response

Dear Reviewer,

Thank you for revising our manuscript entitled “The Use of Different Anthropometric Indices to Assess the Body Composition of Young Women in Relation to the Incidence of Obesity, Sarcopenia and the Premature Mortality Risk”.

We greatly appreciate the time and efforts to review our manuscript and we agree that the proposed changes will contribute to the improvement of our manuscript. We have addressed all issues indicated in reviews, and we believe that the revised version can meet the journal publication requirements.

Please find our responses to the Reviewers comments attached. The changes made in the text are highlighted in red.

Yours Sincerely,

Martina Gažarová

Response to Comments from Reviewer #3

Abstract - Line 15 does not need to start off with a number. Thank you for comment. It was corrected (page 1, line 15).

Introduction - Lines 30 – 43 should be combined as one paragraph. Thank you for your suggestion. It was corrected (page 1, lines 30-41).

Sarcopenic obesity needs to be defined, and its significance needs to be elaborated on. This is the cornerstone of your study, and there is nothing about it in the introduction! Thank you for your suggestion. It was added and defined (page 2, lines 80-90).

Lines 44 – 84 are not written like an introduction. Way too many short paragraphs and independent thoughts. What is the overall focus of the manuscript, and why is this study important? Thank you for your valuable advice and recommendation. We tried to modify it, we believe it will meet your expectations (see page 1-3, lines 30-107).

Methods - In the Participants and Study Design section, what happened to the other 128 individuals who participated in the cross-sectional observational study? Also, is this a secondary analysis? Thank you for your question. This is no secondary analysis. During/after measurements we found via interview and data that one hundred and twenty-eight participants did not satisfy the criteria. The most common reason for exclusion was age exceeding 25 years and non-compliance with the principles of anthropometric measurement – not fasting; excess coffee, alcohol and fat intake less than 8 h prior to testing; physical activity just before the measurement.

Stats - What is the number of participants presented by the power analysis? Thank you for your question. Prior to data collection, the appropriate sample size and power were calculated using G*Power. Based on the result, 70 participants were needed to determine the minimal detectable effect. The number of participants in this study therefore were more than the required sample size.

In the stats section, there is no mention of a predictor model employed. This model is imperative when using independent variables to predict an outcome. Essentially, a regression model needs to be run. Thank you very much for the warning and valuable advice. We performed regression analysis, of course, but ultimately our goal was not to predict outcomes. For these purposes (prediction of the risk of premature death), in our case we used the ABSI z-score with its critical cut-off points based on the methodology of Krakauer and Krakauer (2012). See lines 187-190.

All the tables need to be redone. They are crowded and cumbersome and are hard to follow/interpret. the variables and headings are bad and appear unprofessional and not readable. The symbols used for significance need to be something else besides abcd etc. Thank you very much for the suggestion. We are very sorry that the manuscript for review was not sufficiently edited in the case of tables. We adjusted the data in the tables to a more readable form (reducing the size, changing the direction of the text, etc.). We hope that these corrections will be received and accepted. We hope that the reviewer will accept our way of distinguishing statistical significance using symbols. For these purposes, it is one of the commonly used methods, which is why we chose it.

Discussion - Your main findings need to be presented first. Then each paragraph after that highlights each of your main findings. Thank you for your suggestion, we tried to improve this part (page 11, lines 329-394).

There are too many directions that the discussion is flowing, and there is no focus. Thank you for your suggestion, we tried to improve this part.

The conclusion is too broad, and is hard for the reader to take away the actual significance of the paper. Thank you for your suggestion, we tried to improve this part (page 14, lines 519-547).

Round 2

Reviewer 2 Report

Accept in present form